# Transcriptome Analysis of the Effect of Nickel on Lipid Metabolism in Mouse Kidney

**DOI:** 10.3390/biology13090655

**Published:** 2024-08-24

**Authors:** Jing Zhang, Yahong Gao, Yuewen Li, Dongdong Liu, Wenpeng Sun, Chuncheng Liu, Xiujuan Zhao

**Affiliations:** 1School of Life Science and Technology, Inner Mongolia University of Science & Technology, Baotou 014020, China; zhang520jing1999@163.com (J.Z.); gyh971023@163.com (Y.G.); liimust@163.com (Y.L.); liudd4114@sina.com (D.L.); sun2804780169@163.com (W.S.); 2Inner Mongolia Key Laboratory of Functional Genome Bioinformatics, Baotou 014010, China; 3College of Life Sciences, Inner Mongolia Agricultural University, Hohhot 010018, China

**Keywords:** AMPK signaling pathway, kidney injury, metabolism, nickel chloride, PPAR signaling pathway

## Abstract

**Simple Summary:**

Lipid and glucose metabolism are intimately linked to many disorders, however, further research is needed to understand how they contribute to damage caused by nickel. According to our research, nickel stress increased the levels of blood glucose and lipid indicators. In addition, oxidative stress and inflammation in the kidney were elevated by nickel stress. We identified and examined the differently expressed genes in the kidney caused by nickel stress using RNA sequencing. Nickel suppressed the expression of genes linked to lipid metabolism as well as the AMPK and PPAR signaling pathways in the kidney, according to bioinformatics analysis and experimental verification.

**Abstract:**

Although the human body needs nickel as a trace element, too much nickel exposure can be hazardous. The effects of nickel on cells include inducing oxidative stress, interfering with DNA damage repair, and altering epigenetic modifications. Glucose metabolism and lipid metabolism are closely related to oxidative stress; however, their role in nickel-induced damage needs further study. In Institute of Cancer Research (ICR) mice, our findings indicated that nickel stress increased the levels of blood lipid indicators (triglycerides, high-density lipoprotein, and cholesterol) by about 50%, blood glucose by more than two-fold, and glycated serum protein by nearly 20%. At the same time, nickel stress increased oxidative stress (malondialdehyde) and inflammation (Interleukin 6) by about 30% in the kidney. Based on next-generation sequencing technology, we detected and analyzed differentially expressed genes in the kidney caused by nickel stress. Bioinformatics analysis and experimental verification showed that nickel inhibited the expression of genes related to lipid metabolism and the AMPK and PPAR signaling pathways. The finding that nickel induces kidney injury and inhibits key genes involved in lipid metabolism and the AMPK and PPAR signaling pathways provides a theoretical basis for a deeper understanding of the mechanism of nickel-induced kidney injury.

## 1. Introduction

As a natural element, nickel is widely distributed in soil, water, and air [1]. The amounts of nickel in soil, rivers, and air are about 13–37 mg/kg, 0.7 μg/dm^3^, and 20 ng/m^3^ [2], respectively. Nickel has abundant uses in human society, such as alloy production, electroplating, nickel-cadmium battery production, and catalysts in the chemical and food industries. The widespread use of nickel in industry and food processing increases the risk of human exposure to nickel through air, household products, food, and drinking water [3].

Although nickel is one of the essential trace elements for the human body, excessive exposure to nickel is extremely harmful to human health [4]. Nickel can cause a variety of diseases, such as skin allergies, respiratory diseases, nervous system diseases, and cancer [5]. There are no clear findings regarding safe exposure concentrations of nickel. According to the European Union 1811 testing, children’s articles should not release more than 0.2 mg/cm^2^ of nickel per week [6]. For occupational exposure, water-soluble nickel compounds (Ni ≥ 1 mg/m^3^) have been shown to be associated with excess respiratory cancer risk in workers [7]. In comparison, other data suggest that when levels are restricted to ≤0.1 mg/m^3^, no association with lung cancer occurs [8]. Nickel absorbed by the human body is mainly excreted through urine, and the kidneys are the main target organs for metal toxicity [9]. It has been shown that exposure to highly nickel-containing compounds causes kidney damage in workers [10]. However, there are few in-depth studies on the mechanism of kidney injury caused by nickel exposure.

The concentration of nickel in human plasma is significantly positively correlated with age, while the estimated glomerular filtration rate (eGFR) is significantly negatively correlated with blood nickel level [11,12]. In patients with chronic kidney disease, nickel exposure is one of the causes of progression to end-stage renal disease [13]. However, other studies have found that human plasma concentrations of nickel are not significantly associated with the decline in kidney function measured by eGFR [14].

Studies in animals have shown that nickel can cause or aggravate body damage through inflammation, oxidative stress, and lipid peroxidation [15]. After oral gavage of NiSO_4_ to rats (20 mg per kilogram body weight per day) for 21 days, Ni caused inflammatory response and increased the expression of TNF-α and IL-6 proteins in the kidney [16]. In chick kidneys, nickel may also induce an inflammatory response via activation of the NF-κB signaling pathway and oxidative stress [17,18]. Studies in goldfish and rats have also shown that NiCl_2_ can lead to inhibition of the activities of the renal antioxidant enzymes superoxide dismutase and glutathione reductase, causing lipid oxidative damage in the kidneys [19,20]. Furthermore, in mice kidneys this compound may induce autophagy through AMPK and PI3K/Akt/mTOR signaling pathways [3].

Studies have shown that AMPK can directly or indirectly affect glucose metabolism and lipid metabolism [21]. However, the regulation of glucose metabolism and lipid metabolism in kidney injury is still unclear. In the current study, we found that nickel caused kidney injury and changes in the expression of genes related to lipid metabolism and the AMPK and PPAR signaling pathways. These results help provide a new understanding of the mechanism of nickel-induced kidney injury.

## 2. Materials and Methods

### 2.1. Mice

ICR (Institute of Cancer Research) mice are widely used in toxicology studies. Therefore, ICR mice were selected to study the mechanism of kidney injury caused by nickel. Six-week-old male ICR mice of SPF grade were purchased from Sipeifu Biotechnology Co., Ltd. (Beijing, China). After the mice were transported to our laboratory, they were fed with standard chow for 7 days before the experiment. The feeding environment simulated natural light for a day/night cycle (12 h/12 h), and the temperature was 25 °C. Ad libitum feeding and drinking. The feeding, drinking, and health status of mice were observed daily. The weight changes of mice were measured weekly.

### 2.2. Intraperitoneal Injection of Nickel Chloride

Twenty mice were randomly divided into two groups. Ten mice were intraperitoneally injected with NiCl_2_ daily, and the other ten mice were intraperitoneally injected with the same volume of 0.9% NaCl. The injection dose of NiCl_2_ was 5 μg of NiCl_2_ per gram of body weight every day. The injections lasted for 28 days. This study was approved by the Institutional Animal Care and Use Committee of the Inner Mongolia University of Science and Technology (NMGKJDX-2021-04-02).

### 2.3. Sampling

After continuous injection for 28 days, mice were deprived of feed and water for 16 h. Before sampling, the mice were anesthetized. Mice were anesthetized using 3% pentobarbital sodium at a rate of 1 mL per kg body weight. The mice were sacrificed 2 min after intraperitoneal injection. Blood was collected by enucleation after anesthesia. The mice were then sacrificed by cervical dislocation, and the kidneys were dissected and isolated.

A kidney was put into a 2 mL Eppendorf tube, then immediately immersed in liquid nitrogen, and finally transferred to a −80 °C freezer. Kidneys for HE staining were fixed with 4% paraformaldehyde solution after isolation. The blood samples of mice were left at room temperature for 30 min, and the serum was separated by cryogenic centrifuge. Serum was stored in a −80 °C freezer.

### 2.4. Weighing and Calculation of Relative Organ Weight

The initial body weights of mice were measured and recorded before NiCl_2_ (or NaCl) injection, and then the body weights of mice were measured every 7 days. The body weights of the mice after NiCl_2_ stress for 28 days were used as the final body weight.

Both kidneys were weighed, and relative organ weight was calculated according to the following formula:Relative organ weight (%) = weight of both kidneys (g)/final weight of mouse (g) × 100%

### 2.5. Blood Test

A total cholesterol assay kit (A111-1-1, Nanjing Jiancheng Bioengineering Institute, Nanjing, China), triglyceride assay kit (A110-1-1, Nanjing Jiancheng Bioengineering Institute), high-density lipoprotein cholesterol assay kit (A112-1-1, Nanjing Jiancheng Bioengineering Institute), and low-density lipoprotein cholesterol assay kit (A113-1-1, Nanjing Jiancheng Bioengineering Institute) were used to measure the content of serum lipids.

A glucose assay kit (F006-1-1, Nanjing Jiancheng Bioengineering Institute) and glycosylated serum protein assay kit (a037-2-1, Nanjing Jiancheng Bioengineering Institute) were used to quantify serum glucose levels. Serum urea nitrogen (BUN) levels, which were quantified using a urea assay kit (C013-1-1, Nanjing Jiancheng Bioengineering Institute), assisted in the assessment of kidney injury.

### 2.6. Quantification of Renal Oxidative Stress and Interleukin-6 Protein Levels

A malondialdehyde (MDA) assay kit (A003-1-2, Nanjing Jiancheng Bioengineering Institute) was used to detect the oxidative stress of kidney. Protein levels of Interleukin-6) in the kidney were quantified by ELISA (H007-1-2, Nanjing Jiancheng Bioengineering Institute). Finally, the wet weight of the kidney tissue was used to homogenize the protein expression values.

### 2.7. HE Staining and Glomeruli Cell Counting

HE staining was performed as previously described [22]. The fixed kidneys were dehydrated and paraffin-embedded. Then, the paraffin-embedded kidneys were cut into 4-μm sections and stained with a hematoxylin eosin (HE) staining kit (G1120, Solarbio, Beijing, China) to evaluate renal morphological changes.

For glomeruli cell counting, statistical analysis was conducted on images from three different regions of two samples in each group. Three glomeruli in each image were randomly selected for cell counting.

### 2.8. qRT-PCR

Total RNA from the kidneys was extracted using the RNA extraction reagent RNAiso Plus (9108, Takara, Kyoto, Japan), and the quality of RNA was determined to meet the requirements of subsequent experiments by agarose gel electrophoresis and the OD260/OD280 ratio. RNA was reverse transcribed into cDNA (RR047A, Takara), and gene expression was measured using TB Green^®^ Premix Ex Taq™ II (RR820A, Takara). *Gapdh* was used as a reference gene. The primer sequences were as follows:

*Gapdh*-F 5′-AGGTCGGTGTGAACGGATTTG-3′

*Gapdh*-R 5′-TGTAGACCATGTAGTTGAGGTCA-3′

*Fasn*-F 5′-AAGCAGGCACACACAATGGA-3′

*Fasn*-R 5′-AGTGTTCGTTCCTCGGAGTG-3′

*Apoa_2_*-F 5′-TATGCAGAGCCTGTTCACT-3′

*Apoa_2_*-R 5′-AATCTCTGAGGTCTTGGCCTT-3′

*Apoa_1_*-F 5′-TGTGGATGCGGTCAAAGACA-3′

*Apoa_1_*-R 5′-TCCAGGAGATTCAGGTTCAGC-3′

### 2.9. Western Blot

(a) Protein extraction: proteins were extracted from the kidney using RIPA (R0010, Solarbio) and PMSF. A low temperature was maintained for the samples during the protein extraction process. The BCA protein assay kit (PC0020, Solarbio) was used to determine the protein concentration. (b) Protein denaturation: the conditions for protein denaturation were 99 °C for 10 min. (c) Protein electrophoresis: 30 µg of each sample was loaded into a sample well of an SDS polyacrylamide gel (15%). Proteins were separated by electrophoresis at 120 V for 90 min. (d) Electrophoretic transfer: the gel was cut following electrophoresis in accordance with the target protein’s size. (e) Blotting: the primary antibodies used were AMPKα Rabbit mAb (5831S, CST, Danvers, MA, USA) (1:1000), Phospho-AMPKα Rabbit mAb (2535S, CST) (1:1000), Fatty Acid Synthase Antibody (3189, CST) (1:1000), and Gapdh (14C10) Rabbit mAb (2118T, CST) (1:1000). The secondary antibody used was Goat Anti-Rabbit IgG (H+L) HRP (S0001, affinity, Liyang, China) (1:10,000). (f) Detection: ECL Western Blotting Substrate (32106, Thermo Scientific, Waltham, MA, USA) was utilized to detect signals following hybridization.

### 2.10. Transcriptome Analysis

After continuous injection of NiCl_2_ for 28 days, the mice were anesthetized and sacrificed. The kidney was immersed in liquid nitrogen and finally transferred to a −80 °C freezer. Total RNA was extracted using the mirVana miRNA Isolation Kit (Ambion, Waltham, MA, USA). RNA integrity was evaluated using an Agilent 2100 Bioanalyzer (Agilent Technologies, Santa Clara, CA, USA). The samples with RNA Integrity Number (RIN) ≥ 7 were subjected to subsequent analysis. Transcriptome sequencing was completed by Shanghai OE Biotech Co., Ltd. (Shanghai, China). The libraries were constructed using TruSeq Stranded mRNA LTSample Prep Kit (Illumina, San Diego, CA, USA). The libraries were sequenced on the Illumina sequencing platform (llumina HiSeq X Ten, Illumina) and 150 bp paired-end reads were generated. Raw reads were processed using Trimmomatic. The trimmed reads were mapped to the reference genome using hisat2. Differentially expressed genes were identified using DESeq2.

### 2.11. Data Analysis

#### 2.11.1. Screening of Differentially Expressed Genes

Transcriptome sequencing results were filtered according to mean counts, and only transcripts with mean counts greater than 2 were retained for subsequent analysis. Using the DESeq2 package of R language (RStudio, 4.0.4), the counts of transcripts of each sample were standardized by BaseMean, and the fold change was calculated. Statistical significance was analyzed by negative binomial distribution. The differentially expressed genes were screened according to q-value < 0.05, |log2(FoldChange)| > 1 and coding protein. Venn diagram, cluster analysis, and enrichment analysis were performed using an online analysis platform (Shanghai OE Biotech Co., Ltd., Shanghai, China).

#### 2.11.2. Protein-Protein Interaction (PPI)

PPI was performed using the STRING database with high confidence (confidence > 0.7) [23]. In order to better display the results, the data were then imported into Cytoscape (version 3.9.1). The nodes were arranged clockwise from large to small according to the degree, and the more in the central circle, the higher the degree of the node.

#### 2.11.3. Gene Sets

The mouse lipid metabolism gene set was downloaded from the GSEA database (Systematic name: MM15193). The AMPK signal pathway gene sets were downloaded from the PathCards database of the GeneCardsSuite online platform (https://pathcards.genecards.org/Pathway/1843 (accessed on 10 June 2024)). The PPAR pathway gene sets were downloaded from the GSEA database (Systematic name: MM15995) and the PathCards database of GeneCardsSuite online platform (https://pathcards.genecards.org/Pathway/701 (accessed on 10 June 2024)).

The significance of experimental data was analyzed by Student’s *t*-Tests. Values of *p* < 0.05 were considered statistically significant (* = *p* < 0.05; ** = *p* < 0.01).

## 3. Results

### 3.1. Glucose Metabolism and Lipid Metabolism Were Affected in Mice after NiCl_2_ Stress

Compared with the NaCl-injected mice, the body weights of the NiCl_2_-injected mice were not increased (Figure 1a). Triglyceride (TG), high-density lipoprotein (HDL), and cholesterol (CHO) levels were increased by ~1.5-fold in the mice injected with NiCl_2_ (Figure 1b). Simultaneously, we tested glucose and glycated serum protein (GSP) to evaluate the effect of NiCl_2_ on blood glucose parameters. Serum glucose (GLU) and GSP levels were also significantly increased after NiCl_2_ injection (Figure 1c,d).

Considering the critical role of the kidneys in glucose and lipid metabolism as well as in the storage and excretion of Ni^2+^, we measured serum urea nitrogen (BUN) concentrations as an indicator of kidney function. BUN levels in the serum of NiCl_2_-injected mice showed a trend towards a higher level but this was not statistically significant (Figure 1e).

### 3.2. NiCl_2_ Stress Caused Kidney Injury

Previous studies have found that nickel can cause kidney damage in humans [10]. We therefore examined the kidneys of mice after nickel injection. No obvious abnormalities were observed in the shape and size of the kidneys in mice injected with NaCl or NiCl_2_ (Figure 2a). The calculation of relative kidney weight showed that nickel stress caused kidney hypertrophy (Figure 2b). HE staining of these kidneys showed that NiCl_2_ stress caused glomerular enlargement and increased the number of glomerular cells (Figure 2d).

Levels of both MDA and Interleukin-6 (IL-6) were significantly higher in the kidneys of NiCl_2_-injected mice than in those of their NaCl-injected peers (Figure 2e,f). Collectively, these data suggest that injecting mice with NiCl_2_ for 28 days causes kidney damage. 

### 3.3. Transcriptome Sequencing-Based Analysis of the Effect of NiCl_2_ Stress on Mouse Kidneys

To investigate the molecular mechanism of NiCl_2_-induced kidney injury, mouse kidneys were collected and subjected to transcriptome sequencing. According to the sequencing results, the similarity between samples was analyzed by using the clustering method to calculate the distance between samples according to gene expression (Figure 3a,b). Samples within groups were well clustered in both NaCl-injected mice and NiCl_2_-injected mice (Figure 3a). The heatmap showed that the transcription levels of samples in a given group were similar, and there was a large difference between the groups (Figure 3b). We screened differentially expressed genes by |log2(FoldChange)| > 1 and q-value < 0.05 (Figure 3c). A total of 223 differentially expressed genes were obtained, of which 119 genes were up-regulated and 104 genes were down-regulated (Appendix A).

In order to better describe the functions of differentially expressed genes, GO enrichment analysis was performed for all differentially expressed genes, up-regulated differentially expressed genes, and down-regulated differentially expressed genes (Appendix A), respectively. The GO enrichment analysis results for all differentially expressed genes showed that, in terms of biological process, the differentially expressed genes were mainly related to lipid biosynthesis and metabolism. In terms of cellular components, differentially expressed genes were related to the extracellular matrix, and in terms of molecular function, differentially expressed genes were related to lipid synthesis (Figure 4a).

Compared with the GO enrichment analysis results for all differentially expressed genes, the GO enrichment analysis results for up-regulated differentially expressed genes and down-regulated differentially expressed genes were specific (Figure 4b,c). Regarding the top 30 terms in the GO enrichment analysis of up-regulated differentially expressed genes, multiple terms were related to metabolism (Figure 4b red box). Positive regulation of cholesterol esterification, sterol biosynthetic process, phospholipid binding, glucuronosyl transferase activity, and low-density lipoprotein particle receptor binding appeared in the top 30 terms of GO enrichment analysis for down-regulated differentially expressed genes. In the GO enrichment analysis for all differentially expressed genes, these terms did not rank in the top 30.

The differentially expressed genes caused by NiCl_2_ stress may be involved in the regulation of signaling pathways, so we performed a KEGG enrichment analysis. Total differentially expressed genes were enriched in lipid digestion, and metabolism, glucose metabolism (Figure 5a). These include the AMPK and PPAR signaling pathways, which are closely related to glucose metabolism and lipid metabolism [24,25].

The top 20 pathways were not the same in the enrichment analysis of up-regulated differentially expressed genes and total differentially expressed genes (Figure 5b). The different pathways for up-regulated differentially expressed genes include tryptophan metabolism, FoxO signaling pathway, p53 signaling pathway, Mucin type O-glycan biosynthesis, biosynthesis of unsaturated fatty acids, C-type lectin receptor signaling pathway, cocaine addiction, fatty acid degradation, and leukocyte transendothelial migration.

The enrichment analysis results of down-regulated differentially expressed genes included specific pathways related to fatty acid biosynthesis, steroid biosynthesis, retinol metabolism, caffeine metabolism, and cysteine and methionine metabolism (Figure 5c).

We also undertook a PPI analysis of the 223 differentially expressed genes using the STRING database (Figure 6a). The results showed that 76 of the 223 genes were included in this network (Appendix A). Except for the central *Alb*, the other genes were divided into three levels. According to the degree value, the 11 Hub genes in the center were used for subsequent analysis. Based on the enrichment analysis results, we found that NiCl_2_ stress has important effects on renal lipid metabolism. Therefore, we analyzed these 11 Hub genes with the mouse lipid metabolism gene set in the GSEA database by Venn diagram (Figure 6b). Among these, the intersection of Hub genes and lipid metabolism gene set included 7 genes: *Fasn*, *Apoa1*, *Apoa2*, *Hmgcs2*, *Apoe*, *Alb*, and *Mvk*.

We also analyzed these 11 Hub genes with the AMPK signaling pathway and the PPAR signaling pathway gene sets using Venn diagrams (Figure 6c,d). Venn diagram analysis of the 11 Hub genes and the AMPK signaling pathway gene set identified a single common gene: *Fasn.* These 11 Hub genes also intersected with the gene sets of the PPAR signaling pathway (GSEA database and PathCards database), yielding two common genes: *Apoa1* and *Apoa2*. Therefore, we propose that nickel injection can affect the expression of genes related to lipid metabolism in the kidney.

### 3.4. Effects of Nickel Chloride Stress on Key Genes of the AMPK Signaling Pathway and the PPAR Signaling Pathway

In the Hub genes, *Fasn* (related to lipid metabolism and the AMPK signaling pathway) along with *Apoa1* and *Apoa2* (related to the PPAR signaling pathway) were selected for validation at the mRNA level. *Fasn*, *Apoa1*, and *Apoa2* RNA levels were lower in the NiCl_2_-injected mice than in the NaCl-injected mice (Figure 7a), consistent with the sequencing results.

We also examined Fasn protein levels in NiCl_2_-injected mice and NaCl-injected mice by Western blot analyses. This revealed that the kidneys of NiCl_2_-injected mice contained less Fasn protein than the NaCl-injected mice (Figure 7b, c). AMPKα can inhibit the expression of FASN and reduce triacylglycerol accumulation in hepatocytes [26,27]. Therefore, we reasoned that the NiCl_2_-injections may affect the expression of AMPK in the kidney. Evaluation of the AMPKα subunit of AMPK showed that NiCl_2_-stress decreased AMPKα protein levels in this organ (Figure 7b,d).

## 4. Discussion

We found that nickel chloride can cause oxidative stress, early inflammation, and dysfunction of the kidney. Moreover, that nickel chloride stress leads to dyslipidemia, characterized by increased serum levels of CHO, HDL, and TG. Previous studies have indicated that lipid accumulation in tissues and organs can cause lipotoxicity, and that in the kidney this may lead to injury by inducing oxidative stress, endoplasmic reticulum stress, and mitochondrial dysfunction [28]. The lipid in the glomerulus can stimulate the proliferation of basement membrane cells and increase the volume of the extracellular matrix. At the same time, monocytes and macrophages may infiltrate into the glomerulus to phagocytize lipids. Lipid accumulation in these cells may lead to the formation of foam cells and aggravate glomerular sclerosis [29]. 

Nickel chloride stress also caused changes in blood glucose and glycoprotein. Hyperglycemia can cause diabetic nephropathy [30]. Early studies suggest that inflammation is the key cause of diabetic nephropathy. In fact, glucose can also affect the structure and function of the extracellular matrix through glycosylation and induce oxidative stress to cause kidney injury [31]. The enhanced glycolysis caused by high glucose leads to the accumulation of metabolites such as lactate and pyruvate. The accumulation of these metabolites can cause acidosis and kidney injury [32]. The increase in glycolytic activity of proximal renal tubules is accompanied by an increase of HIF-1α expression, which leads to atrophy of proximal renal tubules and mitochondrial dysfunction [33].

Abnormal glucose metabolism can also affect renal function through lipid metabolism. Glucose can promote the synthesis and accumulation of fatty acids in the body and increase the cholesterol content of the serum. Increased cholesterol will be deposited in non-adipose tissues, including the pancreas, liver, heart, kidney, and vascular [34]. Therefore, renal injury caused by Ni may be related to glucose metabolism and lipid metabolism. Because of the important role of kidneys in metabolic processes, it is important to study the metabolomic response of kidneys to Ni stress.

In addition, we discovered that NiCl_2_ can directly affect metabolism in the kidneys. Nickel stress caused expression changes of several genes including *Apoa1*, *Apoa2*, and *Fasn*. These genes are the key genes of PPAR and AMPK signaling pathways. The PPAR signaling pathway and the AMPK signaling pathway are closely related to glucose and lipid metabolism in the kidney.

## 5. Conclusions

Nickel chloride can cause an increase in the glucose and lipid content of blood, induce oxidative stress and damage to kidney. Transcriptome sequencing data showed that nickel affected the expression of lipid metabolism related genes in the kidney.

The AMPK and PPAR signaling pathways play important roles in glucose and lipid metabolism. Therefore, nickel may regulate AMPK and PPAR signaling pathways and affect metabolism, leading to kidney injury.

## Figures and Tables

**Figure 1 biology-13-00655-f001:**
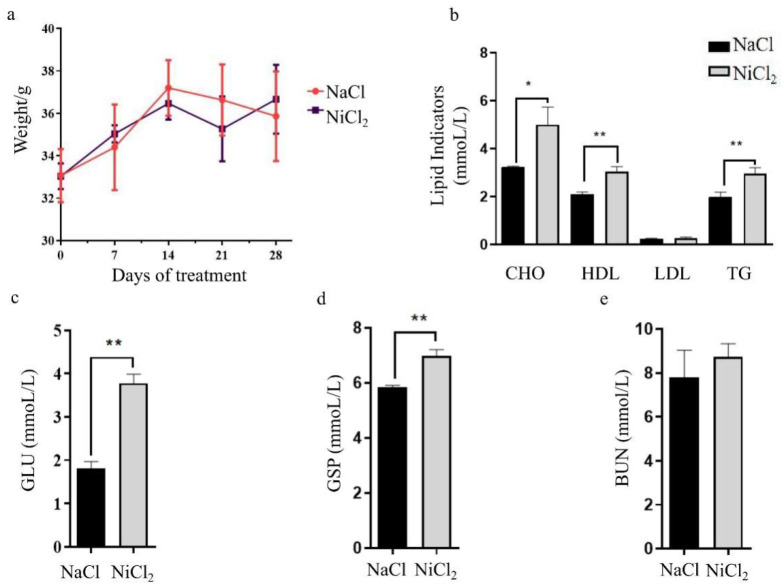
Effect of NiCl_2_ injection on glucose and lipid metabolism in mice. NaCl—mice injected with NaCl. NiCl_2_—mice injected with NiCl_2_. (**a**) The body weights of mice were measured and compared, N = 5. (**b**) After NiCl_2_ stress for 28 days, cholesterol (CHO), high-density lipoprotein (HDL), low density lipoprotein (LDL), and triglyceride (TG) content of serum were measured, N = 5. * = *p* < 0.05, ** = *p* < 0.01. (**c**–**e**) After NiCl_2_ stress for 28 days, blood glucose (GLU), glycated serum protein (GSP), and blood urea nitrogen (BUN) content of serum were measured, N = 5. *** = p* < 0.01.

**Figure 2 biology-13-00655-f002:**
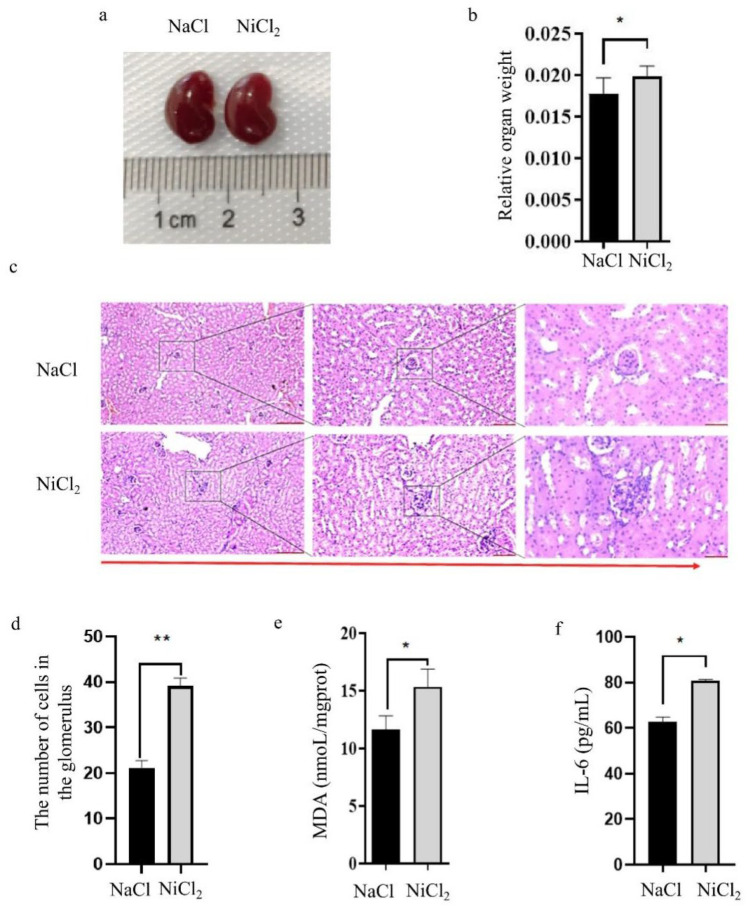
Effects of NiCl_2_ stress on renal function. NaCl—mice injected with NaCl. NiCl_2_—mice injected with NiCl_2_. (**a**) Representative image of the kidneys. (**b**) Relative kidney weights were calculated, N = 5. * = *p* < 0.05. (**c**) HE staining of mouse kidneys. Left panel scale: 250 μm, middle panel scale: 100 μm, and right panel scale: 50 μm. (**d**) Statistics of the number of cells in the glomerulus, N = 18, ** = *p* < 0.05. (**e**,**f**) Determination of malondialdehyde (MDA), and Interleukin 6 (IL-6) content of mouse kidneys, N = 5. * = *p* < 0.05, *** = p* < 0.01.

**Figure 3 biology-13-00655-f003:**
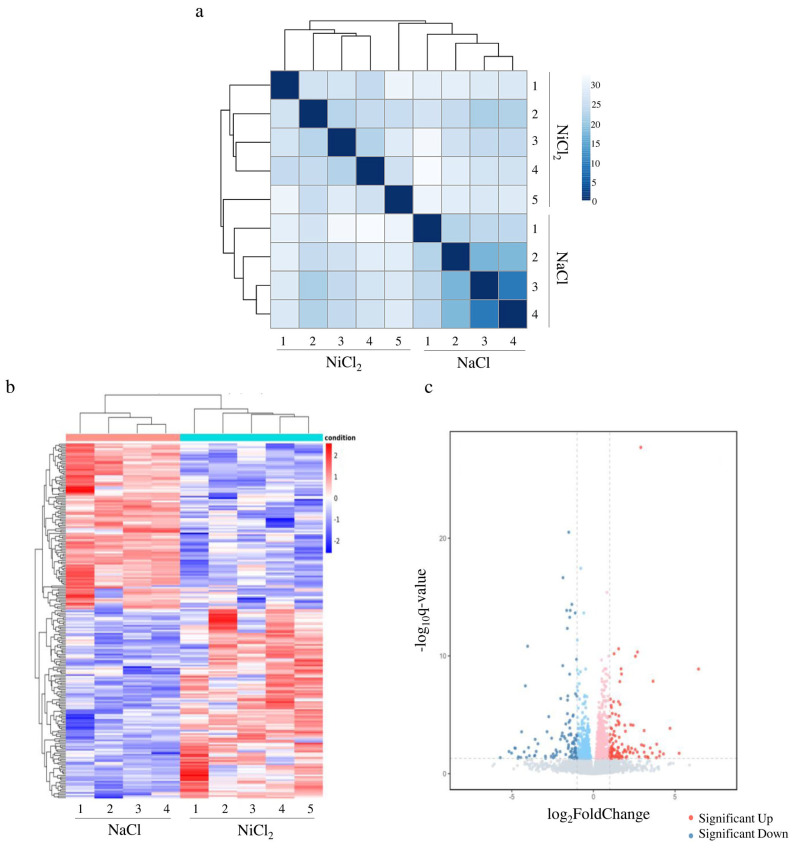
Screening of differentially expressed genes. NaCl—mice injected with NaCl. NiCl_2_—mice injected with NiCl_2_. (**a**) Cluster analysis between samples. (**b**) Hierarchically clustered heatmap of gene expression. (**c**) Volcano map of differentially expressed genes.

**Figure 4 biology-13-00655-f004:**
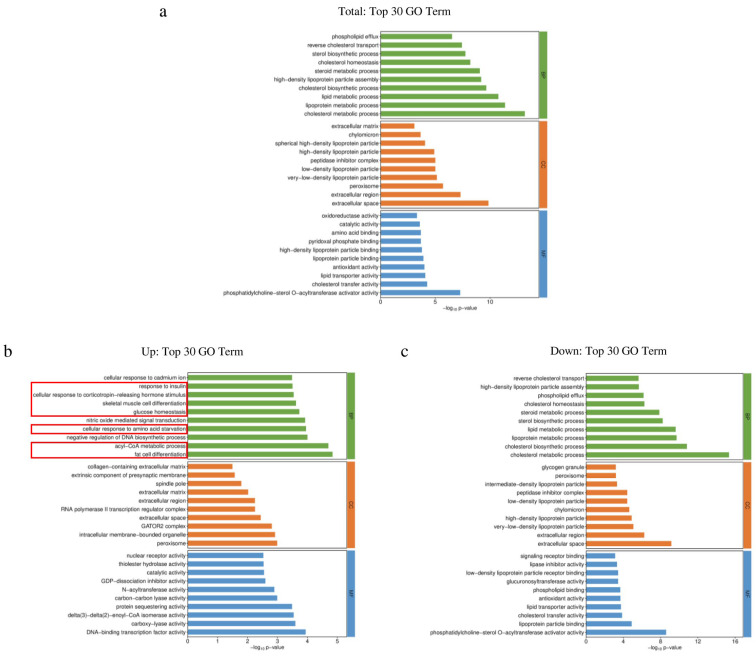
Results of GO enrichment analysis. NaCl—mice injected with NaCl. NiCl_2_—mice injected with NiCl_2_. BP (green)—Biological Process. CC (orange)—Cellular Component. MF (blue)—Molecular Function. (**a**) GO enrichment analysis for all differentially expressed genes. (**b**) GO enrichment analysis for up-regulated differentially expressed genes. (**c**) GO enrichment analysis for down-regulated differentially expressed genes.

**Figure 5 biology-13-00655-f005:**
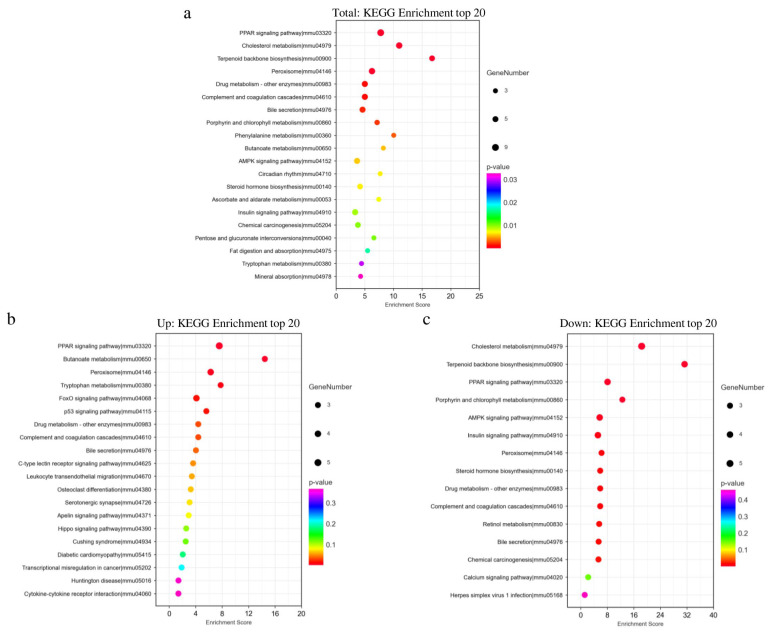
KEGG enrichment analysis. NaCl—mice injected with NaCl. NiCl_2_—mice injected with NiCl_2_. Sorted by *p*-value and listhits (listhits > 2). (**a**) KEGG enrichment analysis of all (total) differentially expressed genes. (**b**) KEGG enrichment analysis of up-regulated differentially expressed genes. (**c**) KEGG enrichment analysis of down-regulated differentially expressed genes.

**Figure 6 biology-13-00655-f006:**
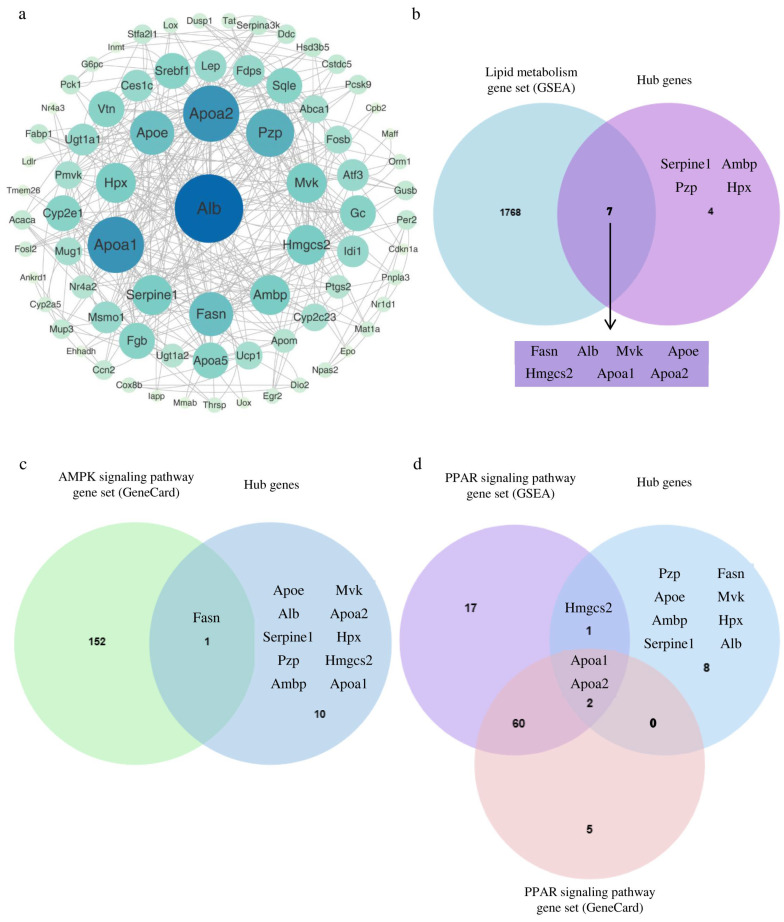
Screening of key genes of lipid metabolism in the kidney under Ni^2+^ stress. (**a**) PPI analysis. (**b**) Venn diagram analysis of Hub genes and lipid metabolism gene set. (**c**) Venn diagram analysis of Hub genes and the AMPK signal pathway gene set. (**d**) Venn diagram analysis of Hub genes and the PPAR signal pathway gene set.

**Figure 7 biology-13-00655-f007:**
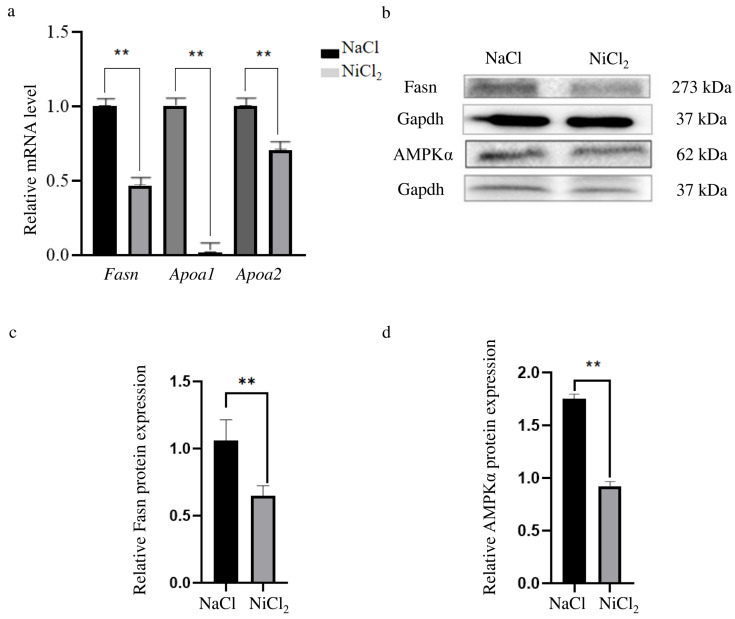
Detection of key genes in the AMPK signaling pathway and the PPAR signaling pathway. NaCl—mice injected with NaCl. NiCl_2_—mice injected with NiCl_2_. (**a**) The expression of *Fasn*, *Apoa1* and *Apoa2* in kidney of mice injected with NaCl or NiCl_2_ for 28 days were detected by qPCR. *Gapdh* was used as a normalized reference. The values represent the mean ± SEM (n = 3). ** = *p* < 0.01. (**b**) Analysis of Fasn and AMPKα protein expression by Western blot. Gapdh protein was used as a normalized reference. (**c**) Quantitative analysis of Fasn protein expression in Figure 7c. N = 4, ** = *p* < 0.05. (**d**) Quantitative analysis of AMPKα protein expression in Figure 7c. N = 3, ** = *p* < 0.05.

## Data Availability

All data generated or analysed during this study are included in this published article.

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
