# Peer review of "Transcriptome Analysis of the Effect of Nickel on Lipid Metabolism in Mouse Kidney"

_biology, 2024, doi:10.3390/biology13090655_

Round 1

Reviewer 1 Report

Comments and Suggestions for Authors

  1. By incorporating below revisions, the manuscript can enhance clarity, completeness, and credibility, thereby increasing reader interest and understanding.
  2. Title: Consider revising to a more informative and concise title.

  3. Abstract: Include quantitative highlights of result, such as significant percentage increases or decreases.

  4. Methods Section: Provide a detailed description of the western blot procedure, specifying how proteins were transferred to the PVDF membrane and the dilutions of primary and secondary antibodies used.

  5. Methods Section (Regarding differential gene expression): Clarify that differentially expressed genes were screened with a q-value < 0.05 and fold chage > 2 initially, as per volcano plot of figure 3  the fold changes > 1 was consider significance?, revised Figure 3 accordingly.

  6. Results Section (Figure 2f): Specify the number of replicates and fields captured for image collection. Consider providing quantitative values for globular cells and include replicates images in supplemental files.

  7. Figure 7b: Include the full western blot image for review purpose only, add information about replicates. Recommend adding quantitative representations of western blot data.

  8. Discussion Section: Update references to include the latest literature that supports and justifies the obtained results.

Author Response

Responds to the reviewer 1’s comments:

  • Title: Consider revising to a more informative and concise title.

Response: Thank you for your suggestion. We have revised the title.

As follows: “Transcriptome analysis of the effect of nickel on lipid metabolism in mouse kidney”.

  • Abstract: Include quantitative highlights of result, such as significant percentage increases or decreases.

Response: Thank you for your opinion. We added significant percentage increases or decreases in the Abstract.

As follows: “the finding indicated that nickel stress increased the levels of blood lipid indicators (triglycerides, high-density lipoprotein, and cholesterol) by about 50%, blood glucose by more than twice, and glycated serum protein by nearly 20%. At the same time, nickel stress increased oxidative stress (malondialdehyde) and inflammation (Interleukin 6) by about 30% in the kidney.”

  • Methods Section: Provide a detailed description of the western blot procedure, specifying how proteins were transferred to the PVDF membrane and the dilutions of primary and secondary antibodies used.

Response: Thank you for your advice. We described the Western blot procedure in the Materials and methods.

As follows: (a) Protein extraction: Proteins were extracted from the kidney using RIPA (R0010, Solarbio) and PMSF. During the protein extraction, keep the sample at a low temperature. The BCA protein assay kit (PC0020, Solarbio) was used to determine the protein concentration. (b) Protein denaturation: The conditions for protein denaturation were 99°C for 10 min. (c) Protein electrophoresis: Load protein samples (30 µg per sample) into the wells of SDS polyacrylamide gel (15%). The conditions for electrophoretic separation of proteins were 120 V for 90 min. (d) Electrophoretic transfer: After electrophoresis, cut the gel according to the size of the target protein. Place a protein-containing polyacrylamide gel in direct contact with the PVDF membrane and "sandwich" this between two electrodes submerged in a conducting solution. When an electric field is applied, the proteins move out of the polyacrylamide gel and onto the surface of the membrane. (e) Blot: The primary antibodies used were AMPKα Rabbit mAb (5831S, CST) (1:1000), Phospho-AMPKα Rabbit mAb (2535S, CST) (1:1000), Fatty Acid Synthase Antibody (3189, CST) (1:1000), and GAPDH (14C10) Rabbit mAb (2118T, CST) (1:1000). The secondary antibody used was Goat Anti-Rabbit IgG (H+L) HRP (S0001, affinity) (1:10000). (f) Detect: After hybridization, ECL Western Blotting Substrate (32106, Thermo Scientific) was used for signal detection.

  • Methods Section (Regarding differential gene expression): Clarify that differentially expressed genes were screened with a q-value < 0.05 and fold chage > 2 initially, as per volcano plot of figure 3  the fold changes > 1 was consider significance?, revised Figure 3 accordingly.

Response: Thank you for your careful work. Our description in Materials and methods was non-standardized. I am so sorry for this typo; we have corrected this problem in revised manuscript.

As follows:

"Materials and methods

12.Data analysis

The differentially expressed genes were screened according to q-value<0.05, |log2(FoldChange)|>1 and coding protein."

  • Results Section (Figure 2f): Specify the number of replicates and fields captured for image collection. Consider providing quantitative values for globular cells and include replicates images in supplemental files.

Response: Thank you for your suggestion. Statistics on the number of cells in the glomeruli are helpful to better illustrate the effect of NiCl2 on the kidney. Statistical analysis was conducted on images from three different regions of two samples in each group. Three glomeruli in each image were randomly selected for cell counting.

The statistical process of cell counting was added to the Materials and methods, and the results of cell counting were added to Figure 2.

As follows:

"Materials and methods

8.HE staining

Glomeruli cell counting. Statistical analysis was conducted on images from three different regions of two samples in each group. Three glomeruli in each image were randomly selected for cell counting."

(g) Statistics of the number of cells in the glomerulus., N=18, **: p<0.05.

  • Figure 7b: Include the full western blot image for review purpose only, add information about replicates. Recommend adding quantitative representations of western blot data.

Response: Thank you for your advice. Quantification of Western blot can better describe protein expression. Quantification of Western blot was added to Figure 7. At the same time, we provided full Western blot image.

As follows:

(c-d) Quantitative analysis of AMPKα and Fasn protein expression in Figure 7(b). The reference protein was Gapdh. N=3, **: p < 0.05.

  • Discussion Section: Update references to include the latest literature that supports and justifies the obtained results.

Response: Thank you for your opinion, we updated the references.

As follows:

[22] XIANG Q, XIN L, LANGHUI L, et al. The role of perirenal adipose tissue deposition in chronic kidney disease progression: Mechanisms and therapeutic implications [J]. Life Sci, 2024, 352(0).doi.org/10.1016/j.lfs.2024.122866

[23] LEE L E, DOKE T, MUKHI D, et al. The key role of altered tubule cell lipid metabolism in kidney disease development [J]. Kidney Int, 2024, 106(1): 24-34.doi.org/10.1016/j.kint.2024.02.025

[25] JI R, CHEN W, WANG Y, et al. The Warburg Effect Promotes Mitochondrial Injury Regulated by Uncoupling Protein-2 in Septic Acute Kidney Injury [J]. Shock, 2021, 55(5): 640-8.doi.org/10.1097/SHK.0000000000001576

Reviewer 2 Report

Comments and Suggestions for Authors

Subject of paper is the effect of nickel chloride stress on lipid metabolism in mouse kidney.

Authors should state which anestethic for animals was used, in which dosage, how long was anesthesia duration previous to sacrificing. It is necessary to have parameters measured before anesthesing to check is there influence of anesthetic on blood count and biochemical parameters and oxidative stress parameters.

Authors should rephrase kit used for RNA extraction simmilar to antibodes previously stated. Why was GAPDH used as reference gene? Was this tested in treated versus non treated animals? Was expression of refrerence gene stable in both? Authors should rephrase sentence regarding protein quantity loaded on PAA gel.

It is not stated in what conditions are genes of interest differentially expressed. Authors should state by which method was transcriptome screening performed and include this in materials and methods section. Also, the significance of examining protein-protein interaction is not clear, since it is mentioned in data analysis section, but it is not referenced.

English language should be rechecked and corrected.

Comments on the Quality of English Language

English language should be corrected under figure legends.

Author Response

Responds to the reviewer 2’s comments:

  1. Authors should state which anestethic for animals was used, in which dosage, how long was anesthesia duration previous to sacrificing. It is necessary to have parameters measured before anesthesing to check is there influence of anesthetic on blood count and biochemical parameters and oxidative stress parameters.

Response: Thank you for your suggestion. Mice were anesthetized using 3% sodium pentobarbital 1 mL per kg body weight. The mice were sacrificed 2 min after intraperitoneal injection. The above content was added to the Materials and methods. As follows:

“Mice were anesthetized using 3% pentobarbital sodium 1 mL per kg body weight. The mice were sacrificed 2 min after intraperitoneal injection.”

To detect the effect of pentobarbital sodium on various parameters is a very good suggestion and a very noteworthy problem. Pentobarbital sodium is widely used in the anesthesia of laboratory animals[1], and it has been found that pentobarbital sodium does not cause changes in blood chemistry parameters[2]. At the same time, mice in each group were injected with pentobarbital sodium. So various parameters were not tested in this experiment.

  1. Authors should rephrase kit used for RNA extraction simmilar to antibodes previously stated. Why was GAPDH used as reference gene? Was this tested in treated versus non treated animals? Was expression of refrerence gene stable in both? Authors should rephrase sentence regarding protein quantity loaded on PAA gel.

Response: Thank you for your comment. Gapdh (Glyceraldehyde-3-phosphate dehydrogenase) is commonly used as a reference gene in gene expression studies. Gapdh is expressed at relatively constant levels in most cell types and under different experimental conditions, making it a suitable reference gene for normalizing gene expression data and ensuring accurate and reliable results. In previous experiments, we compared the expression of Gapdh, β-actin, and Tubulin at mRNA level. The expression of these genes were relatively stable under Ni stress.

We rephrased the sentence regarding protein quantity loaded on PAA gel. As follows:

“Protein electrophoresis: Load protein samples (30 µg per sample) into the wells of SDS polyacrylamide gel (15%). The conditions for electrophoretic separation of proteins were 120 V for 90 min.”

  1. It is not stated in what conditions are genes of interest differentially expressed. Authors should state by which method was transcriptome screening performed and include this in materials and methods section. Also, the significance of examining protein-protein interaction is not clear, since it is mentioned in data analysis section, but it is not referenced.

Response: Thank you for your careful work. In this study, kidney tissues were sequenced after 28 days of NiCl2 treatment, and differentially expressed genes were analyzed. Based on your suggestion, we added transcriptome analysis to the Materials and methods and refined protein-protein interaction detection.

As follows:

Transcriptome analysis

After continuous injection of NiCl2 for 28 days, the mice were anesthetized and sacrificed. The kidney was immersed in liquid nitrogen, and finally transferred to the -80℃ ultra-low temperature refrigerator. Transcriptome sequencing was completed by Shanghai OE Biotech Co., Ltd. (China).

Protein-protein interaction detection

Protein-protein interaction analysis was performed using the STRING database with high confidence (confidence>0.7). Set PPI enrichment p-value<1.0e-16 and outputted data. In order to better display the results, the data were then imported into Cytoscape (version 3.9.1).

5.English language should be rechecked and corrected.

Response: Thank you for your advice. We rechecked and corrected the manuscript and made corrections to several sentences, such as:

“Bioinformatics analysis and experimental verification showed that nickel inhibited the expression of genes related to lipid metabolism and the AMPK and PPAR signaling pathways in the kidney.”

“In the GO enrichment analysis of all differentially expressed genes, these terms did not rank in the top 30.”

“According to the degree value, the 11 Hub genes in the center were used for subsequent analysis.”

  1. Laferriere, C.A. and D.S.J. Pang, Review of Intraperitoneal Injection of Sodium Pentobarbital as a Method of Euthanasia in Laboratory Rodents. J Am Assoc Lab Anim Sci, 2020. 59(3): p. 343.
  2. Tsubokura, Y., et al., Effects of pentobarbital, isoflurane, or medetomidine-midazolam-butorphanol anesthesia on bronchoalveolar lavage fluid and blood chemistry in rats. J Toxicol Sci, 2016. 41(5): p. 595-604.

Reviewer 3 Report

Comments and Suggestions for Authors

Generally, the manuscript is interesting and has a vital path of investigating the toxic effects of the excessed dose of nickel in mice models. By sensibly reviewing the manuscript, I found that all parts of the manuscript were written in a well-scientific manner, and the experiments were performed professionally; the results have been summarized and discussed satisfactorily. Otherwise, some comments need to be considered before the manuscript is accepted to be published in the Journal of Biology; I hope that will help to improve the scientific value of the manuscript.

1.   Why did the authors use the ICR mice?, mention the full expression of ICR in first time.

2.   In the abstract, please change the “we found” to any other expression such as “the finding indicated that…..”

3.   In the abstract, please rewrite this sentence “At the same time, nickel stress caused oxidative stress (MDA, GSP-Px) and inflammation (IL-6) in the kidney” to clarify what happened in the mentioned biomarker.

4.   In the abstract, rewrite the sentences” and had an effect on the AMPK and PPAR signaling pathways” to clarify what type of effects happen in the AMPK and PPAR signaling pathways.

5.   In the abstract, revise this sentence “The finding that nickel affects renal metabolism can provide a theoretical basis for a deeper understanding of the mechanism of nickel-induced kidney injury” to conclude the main findings.

6.   Please mention the full names of all abbreviations mentioned in the abstract.

7.   Key words, should be written in alphabetical order.

8.   In comparison, Please mention the safe exposure level of nickel for humans, especially children.

9.   Please mention the detection levels of nickel in environmental elements (from the literature).

10. The injection dose of NiCl2 was 5 μg of NiCl2 per gram of body weight every day, why did the authors choose this dose and duration of exposure?

11. “Or the isolated kidney was immediately washed with PBS and put into a 4% paraformaldehyde solution” This needs to be revised.

12. In  4. Weighing and calculation of relative organ weight, please remove subtitles.

13. The total RNA of the kidney was extracted (9108, Takara), revise this sentence.

14. p<0.05 *, p<0.01, make them italic form in whole manuscript.

15. Mention the full names of biomarkers in captions of figures.

16. Mention the Transcriptome analysis in the materials and methods section before data analysis>

17. In discussion, please rephrase “we had also found that”.

Comments on the Quality of English Language

need slightly improvement 

Author Response

Responds to the reviewer 3’s comments:

  1. Why did the authors use the ICR mice?, mention the full expression of ICR in first time.1.

Response: Thank you for your careful work. ICR mice are widely used in toxicology studies. Therefore, ICR mice were selected to study the mechanism of kidney injury caused by NiCl2. In the Materials and methods, we added a description of the reasons for selecting ICR mice. At the same time, the full expression of ICR has been added in first time.

As follows:

"Materials and methods

  1. Mice

ICR (Institute of Cancer Research) mice are widely used in toxicology studies. Therefore, ICR mice were selected to study the mechanism of kidney injury caused by nickel."

  1. In the abstract, please change the “we found” to any other expression such as “the finding indicated that…..”

Response: Based on the reviewer's suggestion, we revised this sentences in the abstract. As follows:

As follows:

“the finding indicated that nickel stress increased the levels of blood lipid indicators (triglycerides, high-density lipoprotein, and cholesterol) by about 50%, blood glucose by more than twice, and glycated serum protein by nearly 20%. At the same time, nickel stress increased oxidative stress (malondialdehyde) and inflammation (Interleukin 6) by about 30% in the kidney.”

  1. In the abstract, please rewrite this sentence “At the same time, nickel stress caused oxidative stress (MDA, GSP-Px) and inflammation (IL-6) in the kidney” to clarify what happened in the mentioned biomarker.

Response: Thank you for your advice, we revised this sentence in the abstract.

As follows:

“At the same time, nickel stress increased oxidative stress (malondialdehyde) and inflammation (Interleukin 6) by about 30% in the kidney.”

  1. In the abstract, rewrite the sentences” and had an effect on the AMPK and PPAR signaling pathways” to clarify what type of effects happen in the AMPK and PPAR signaling pathways.

Response: Thank you for your advice. We amended this sentence.

As follows:

“Bioinformatics analysis and experimental verification showed that nickel inhibited the expression of genes related to lipid metabolism and the AMPK and PPAR signaling pathways in the kidney.”

  1. In the abstract, revise this sentence “The finding that nickel affects renal metabolism can provide a theoretical basis for a deeper understanding of the mechanism of nickel-induced kidney injury” to conclude the main findings.

Response: Thank you for your careful work. To make our conclusion more accurate, we amended this conclusion as “The finding that nickel induced kidney injury and inhibited the key genes of lipid metabolism and the AMPK and PPAR signaling pathways can provide a theoretical basis for a deeper understanding of the mechanism of nickel-induced kidney injury.”

  1. Please mention the full names of all abbreviations mentioned in the abstract.

Response: Thank you for your advice. The full names of all abbreviations have been mentioned in the abstract, for instance, triglyceride, high-density lipoprotein, glycated serum protein, and so on.

  1. Key words, should be written in alphabetical order.

Response: Thank you for your careful work, we amended the Key words.

As follows: Keywords: AMPK signaling pathway; Kidney injury; Metabolism; Nickel chloride; PPAR signaling pathway

  1. In comparison, Please mention the safe exposure level of nickel for humans, especially children.

Response: Thank you for your careful work. We added the safe exposure level of nickel for humans to the Introduction.

As follows:

“Therefore, according to the European Union 1811 testing, children's articles should not release nickel more than 0.2 mg/cm2 per week[1]. For occupational exposure, water-soluble nickel compounds (Ni≥1 mg/m3) were associated with excess respiratory cancer risk in workers[2]. When Ni≤0.1 mg/m3, there was no correlation with tumorigenesis[3].”

  1. Please mention the detection levels of nickel in environmental elements (from the literature).

Response: Thank you for your careful work. We added the detection levels of nickel in environment to the Introduction.

As follows:

“The amount of nickel in soil, river and air is about 13-37 mg/kg, 0.7 μg/dm3, and 20 ng/m3[4].”

  1. The injection dose of NiCl2 was 5 μg of NiCl2 per gram of body weight every day, why did the authors choose this dose and duration of exposure?

Response: Thank you for your comment. With reference to the published manuscript[5-7], multiple injection doses (2.5 μg, 5 μg, and 10 μg NiCl2 per g body weight) and time points (7, 14, 28, and 56 days) were included in the preliminary study. High concentrations of NiCl2 (10 μg NiCl2 per g body weight) could cause serious damage to multiple organs. The moderate injection dose (5 μg NiCl2 per g body weight) for 28 days had no significant effect on the body weight and the relative weight of various organs. We therefore chose the present injection dose (5 μg of NiCl2 per g body weight every day) for 28 days.

  1. “Or the isolated kidney was immediately washed with PBS and put into a 4% paraformaldehyde solution” This needs to be revised.

Response: Thank you for your advice, we rephrased this sentence.

As follows:

“Kidneys for HE staining were fixed with 4% paraformaldehyde solution after isolated.”

  1. In  4. Weighing and calculation of relative organ weight, please remove subtitles.

Response: Thank you for your comment. we made revisions to the manuscript.

  1. The total RNA of the kidney was extracted (9108, Takara), revise this sentence.

Response: Thank you for your comment, we rephrased this sentence.

As follows:

“Total RNA from the kidneys was extracted using the RNA extraction reagent RNAiso Plus (9108, Takara)”

  1. p<05 *, p<0.01, make them italic form in whole manuscript.

Response: Thank you for your suggestion, we made revisions in whole manuscript.

  1. Mention the full names of biomarkers in captions of figures.

Response: Thank you for your careful work, we added the full name of biomarkers in captions of figures.

As follows:

Figure 1 Effect of NiCl2 injection on glucose and lipid metabolism in mice. NaCl: mice were injected with NaCl. NiCl2: mice were injected with NiCl2. (a) The weight of mice was measured and compared. N=5. (b) After NiCl2 stress for 28 days, cholesterol (CHO), high-density lipoprotein (HDL), low density lipoprotein (LDL), and triglycerides (TG) contents in serum were measured. N=5. p<0.05 *, p<0.01 **. (c-e) After NiCl2 stress for 28 days, blood glucose (GLU), glycated serum protein (GSP), and Blood Urea Nitrogen (BUN) contents in serum were measured. N=5. p<0.01**.

  1. Mention the Transcriptome analysis in the materials and methods section before data analysis.

Response: Thank you for your careful work. Describing Transcriptome analysis before Data analysis has improved our experimental method. We added transcriptome analysis to the Materials and methods.

As follows:

Transcriptome analysis

After continuous injection of NiCl2 for 28 days, the mice were anesthetized and sacrificed. The kidney was immersed in liquid nitrogen, and finally transferred to the -80℃ ultra-low temperature refrigerator. Transcriptome sequencing was completed by Shanghai OE Biotech Co., Ltd. (China).

  1. In discussion, please rephrase “we had also found that”.

Response: Thank you for your advice, we rephrased this sentence.

As follows:

“In addition, we discovered that NiCl2 could directly affect metabolism in the kidneys.”

  1. Silverberg, N.B., et al., Nickel Allergic Contact Dermatitis: Identification, Treatment, and Prevention. Pediatrics, 2020. 145(5).
  2. Buxton, S., et al., Concise Review of Nickel Human Health Toxicology and Ecotoxicology. Inorganics, 2019. 7(7).
  3. Oller, A.R., G. Oberdorster, and S.K. Seilkop, Derivation of PM10 size-selected human equivalent concentrations of inhaled nickel based on cancer and non-cancer effects on the respiratory tract. Inhal Toxicol, 2014. 26(9): p. 559-78.
  4. Harasim, P. and T. Filipek, Nickel in the Environment. Journal of Elementology, 2015. 20(2): p. 525-534.
  5. Yin, H., et al., Nickel induces autophagy via PI3K/AKT/mTOR and AMPK pathways in mouse kidney. Ecotoxicol Environ Saf, 2021. 223: p. 112583.
  6. Xu, L., et al., Role of Txnrd3 in NiCl(2)-induced kidney cell apoptosis in mice: Potential therapeutic effect of melatonin. Ecotoxicol Environ Saf, 2023. 265: p. 115521.
  7. Guo, H., et al., Oxidative stress-mediated apoptosis and autophagy involved in Ni-induced nephrotoxicity in the mice. Ecotoxicol Environ Saf, 2021. 228: p. 112954.

Round 2

Reviewer 2 Report

Comments and Suggestions for Authors

Thanks for acknowledging suggestions, proper corrections were made. 

Author Response

Thank you for your help.